# The Research of 30 mm Detecting Distance of Testing Device for Wire Rope Based on Open Magnetizer

**Mengqi Liu [1], Chi Zhang [1], Xiaoyuan Jiang [1], Yanhua Sun [1,\*], Xiaotian Jiang [1], Ran Li [2,3] and Lingsong He [1]**

1   School of Mechanical Science & Engineering, Huazhong University of Science and Technology, Wuhan 430074, China; d202080231@hust.edu.cn (M.L.); q760931845@163.com (C.Z.); 15671565817@163.com (X.J.); noame@126.com (X.J.); helingsong@hust.edu.cn (L.H.)
2   School of Transportation and Logistics Engineering, Wuhan University of Technology, Wuhan 430063, China; rangly@126.com
3   Three Gorges Navigation Authority, Yichang 443002, China
\*   Correspondence: yhsun@hust.edu.cn

**Abstract:** Wire rope will have defects such as local faults (LF) and loss of metal area (LMA) during the long-term using process. The nondestructive testing method of magnetic flux leakage (MFL) has been widely used in wire rope defect detection. Currently, the detecting distance between magnetic sensors and wire rope with the MFL method is relatively small (2–5 mm). Considering the inner surface of the sensor head is close to the wire rope, it quickly leads to the sensor head scraping off the surface oil of the wire rope or being stuck by a cut wire in the course of MFL detection. Therefore, it is challenging to realize the sensor with MFL detection of wire rope obtaining the weak signal at a large lift-off (>30 mm). This study used finite element analysis to verify if the MFL signal exists at the large lift-off (>30 mm). Meanwhile, the sensor head was improved using an open magnetizer to make the wire rope reach saturation and excite enough magnetic leakage field. By combining magnetic sensing and coupling and a weak analog signal processing method, not only was the signal effectively detected, but also the signal-to-noise ratio (SNR) was improved. Finally, experiments verify the feasibility of detecting defects at a large distance. The method also has been applied in the high-speed detection of wire rope, which can detect broken wire of 1 mm diameter.

**Keywords:** magnetic flux leakage (MFL); large lift-off; coil array coupling; analog signal processing; high-speed; signal-to-noise ratio (SNR)

## 1. Introduction

Wire rope is a slender and vulnerable ferromagnetic component that bears the most load and is related to the safety of life and property. It is widely used in mining, oil, elevators, cableway, transportation, ports, and several other vital fields due to its fatigue resistance and high tensile strength [1]. However, wire rope is susceptible to defects such as broken wire, corrosion, wear, and fatigue during the long-term use process. What is worse, these defects could occur in the internal of the wire rope, which cannot be found using the visual inspection of traditional methods [2,3]. There are growing appeals for effective wire rope detection.

With the continuous improvement of safety awareness, wire rope inspection has been used in many scenes. The practical goal in lots of applications is to realize the safe use of wire rope and avoid wastage caused by prematurely being replaced [4,5]. At present, most scenes have to take the measure of regular replacement of wire rope, which not only contributes to much pressure on resources but also cannot eliminate the risk of sudden fracture. To realize the real-time performance of wire rope inspection without affecting normal production activities, wire rope inspection must develop from off-line handheld to in-service inspection. In-service detection equipment runs under high load for 24 h [6].

The existing methods of detecting wire rope include ultrasonic [7], visual [8], X-ray, electromagnetic method [9], magnetostriction, and so on. The electromagnetic method is the most commonly used in practical applications, and it contains magnetic flux leakage (MFL) methods [10], main flux methods [11], eddy current [12,13], alternating current field measurement, and pulsed eddy current [14]. Among these kinds of methods, the MFL method has a significant effect in addition to simple theory and easy implementation in practical applications. The damage can be determined by measuring the MFL signal [15].

Until now, using the MFL method, the existing detection lift-off distance is very small (2 mm–5 mm) [16]. LEE and Hwang proposed an equation for determining the crack volume when the lift-off is known using the 1/4 RMS algorithm to detect a far-side crack. Besides, the lift-off distance in their study can reach 11 mm [17]. M. Dutta et al. observed the MFL field of a defect varies drastically with lift-off by simulations, and its lift-off distance reaches 10 mm [18]. Wu et al. introduced the defect quantization errors caused by mechanical vibration and electromagnetic noises to analyze the influence of lift-off values and electromagnetic noises [19]. A lift-off-tolerant MFL sensor is developed and tested by Wu et al. based on the magnetic field focusing effect of ferrite cores. It showed high sensitivity at a lift-off distance of 5.0 mm [20]. Azad and Kim acquired the induced current of the coil sensor and optimum values of parameters in magnetizing and sensing units numerically. Finally, the 4.8 mm lift-off distance of the coil sensor was verified [21]. Combined with the thickness of magnetic sensitive elements, the inner surface of the sensor head is usually directly attached to the detected steel wire rope. As a result, the magnetic flux leakage sensor will scrape off the oil stain on the steel wire rope's surface. Therefore, it is crucial to realize that the magnetic flux leakage sensor can detect wire rope defects at a large lift-off distance.

However, with the increasing distance, the field strength of the leakage magnetic field decreases exponentially. Even if the magnetization is enhanced by increasing the number of magnetizers, there is still a problem of weak damage signal overwhelmed by the noise interference at a large lift-off distance, such as strand signal, swing signal, and background noise signal [22]. The frequency of wire rope damage signal is affected by many factors, such as damage type, size, and running speed of wire rope, making signal processing and recognition difficult [23].

As far as we know, little previous research has investigated the large lift-off (>30 mm) detection approach in MFL nondestructive field. Aiming at the challenges and problems mentioned above, the requirements of the sensor head, which can detect wire rope defects at a large distance (>30 mm) with the MFL detection method, are proposed: High sensitivity; High stability; High signal-to-noise ratio (SNR); High amplitude. A novel 30 mm detecting distance testing device for wire rope based on an open magnetizer is proposed in this article. By comparing traditional small lift-off detection, it can be deduced that if a large lift-off detection method can be obtained, weak defect signals can be collected when the magnetizer, sensors, and signal processing system are improved.

In this paper, verification of the proposed method is further investigated by established 3D finite element analysis and relevant pre-test. Aiming at the related difficulties of detecting steel wire rope with a magnetic flux leakage sensor in large lift-off (>30 mm), a magnetic sensing and coupling method of the weak signal is proposed by combining simulation and a large number of experiments. In the experiments, the characteristics and parameters of magnetic sensitive elements are analyzed and compared, and their array arrangements are reasonable. Firstly, the noise such as strand signal is differentially processed at the extraction of magnetic leakage field, which improves the SNR of the signal source, reducing the difficulty of subsequent signal processing and indirectly improving the sensitivity of sensitive elements. Secondly, the appropriate frequency response space is designed by using the analog signal processing method of weak signals. Finally, the detection system is tested, and the test shows that the defects of steel wire rope can be detected in large lift-off (>30 mm). Besides, the magnetic sensing and coupling method and the analog signal processing method of the weak signal are expanded to other applications of minor

damage detection. This article is organized as follows. Section 2 of this paper will model and simulate magnetizer sensor detection at a large lift-off distance, and a preliminary experiment verifies the feasibility. Section 3 analyses the induction coils from 4 different perspectives, including type, array, array coupling, and structure parameter. Then the analog circuit design and circuit simulation are performed in Section 4. Furthermore, the test bench is constructed and the MFL detection system is assembled to start an experiment. The experiments compare different coil arrays and coil structure parameters mentioned in Section 3 to determine the best type of coil sensors, and accompanied by soft magnetic material, permalloy in Section 5. Finally, a discussion and conclusion about the advantages, limitations, and what to do in the future are presented in Sections 6 and 7.

## 2. Magnetizer Modeling and Simulation Analysis

### 2.1. Open Permanent Magnetizer

The principle of MFL detection consists of exposing the wire rope to a strong magnetic field in the axial direction. The wire rope has a higher magnetic permeability than the surrounding air, which increases the density of magnetic lines of flux flowing through the wire rope to those flowing outside. Hence, a surface-breaking defect presents high reluctance to the flow of magnetic lines of flux, thereby causing the flux lines to leak out from the wire rope around the defect. This magnetic flux leakage field is postulated to contain information about the shape and size of the defect, as well as the permeability of the wire rope and the strength of the applied axial field. MFL signals are the MFL field values measured by magnetic sensors placed over the wire rope close to the defect. In practice, the MFL technique is more effective if the applied magnetic field is strong enough to saturate the wire rope [18]. In the MFL method, magnetic sensors such as Hall effect sensors, giant magneto resistances, and induction coils are used to collect MFL distribution from the wire rope's defect. The MFL signal is converted into the electric signal by magnetic sensors and it is convenient to process in the corresponding computer software. Induction coils used Faraday law, and it consists of copper. The Hall sensor is based on the hall effect. As the magnetoresistive (MR) effect appears, various MR sensors have been proposed, such as the giant magnetoresistance sensor, tunneling magnetoresistance, and anisotropic magnetoresistance sensor. A MR sensor is composed of a first and a second thin film layer of magnetic material separated by a layer of non-magnetic metallic material. Magnetizing is the primary step in the magnetic flux leakage testing method. The existing ways of magnetizing wire rope are divided into two types: one is based on the principle of electromagnets to magnetize the wire rope with an energized excitation coil, and the other uses a permanent magnet magnetizing steel wire rope connected by the yoke with high permeability. As shown in Figure 1a, to generalize a stable straight-line magnetic field, the exciting coil should be supplied with strong power and high stability. In this way, exciting coils with a large number of turns and big volumes are required to increase the magnetic field. For this reason, it will take too much time and manpower to wind the exciting coil to encircle a new wire rope [24]. It is also extremely inconvenient when applied in practical working sites. According to the second method, yoke type magnetizer containing two permanent magnets and a high permeability yoke with the detected wire rope compose a magnetic bridge circuit, as shown in Figure 1b. Six sets of the yoke type magnetizer arranged in the circumference direction of the wire rope compose an integral magnetizer. However, for the effect of magnetizing, a small lift-off distance has to consider as an important factor that could rub magnetic sensors which are placed between the wire rope and the yoke, inside the integral magnetizer, resulting in an abnormal signal with anomaly noise.

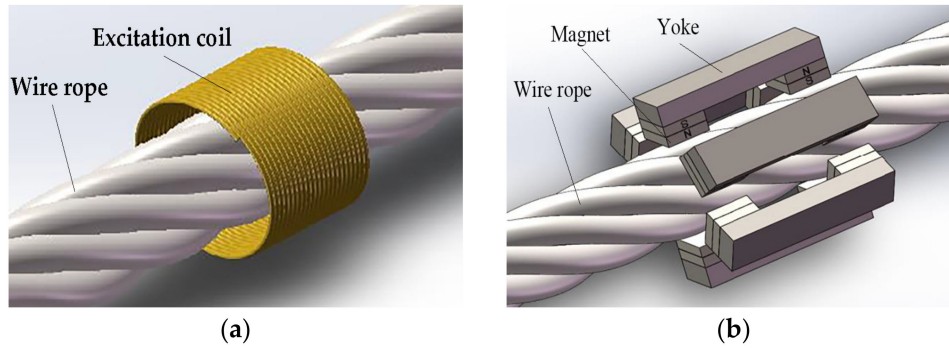

**Figure 1.** (**a**) Exciting coil magnetizer; (**b**) Yoke type magnetizer.

Considering the above two methods, an open permanent magnetizer is proposed, shown in Figure 2. The ring permanent magnets are magnetized along the wire rope axial direction; by using the ring permanent magnets, induction lines can enter into the detected wire rope evenly. Meanwhile, an open structure consisting of 2 semicircular magnets has been designed to encircle the detected wire rope from the middle position by opening the 2 semicircular magnets. Therefore, the symmetric ring structure magnetizer magnetizes the detected wire rope evenly and makes a large lift-off distance detection possible. Once a large lift-off distance is applied in the detecting procedure, the eccentric problem of the wire rope caused by magnets' attraction can be eliminated [25]. The magnetic flux lines distribute more evenly under open magnetizer conditions than under yoke type magnetizer [6].

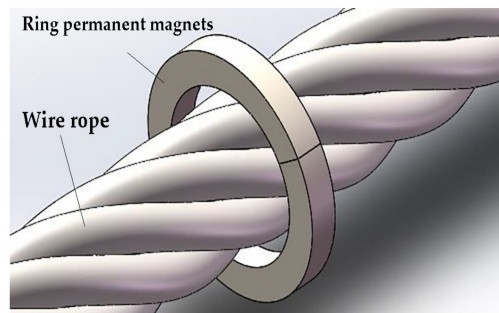

**Figure 2.** Open permanent magnetizing methods.

### 2.2. Analysis of the Feasibility of Large Lift-Off

The finite element method was adopted to simulate the MFL signal of the defected wire rope at a large lift-off distance. The solution type was magnetostatic. The diameter of the wire rope was set to 22 mm. Its material was X52, steel used for oil and gas pipeline transmissions, and a cut was set in the middle part of the wire rope as the break whose length was, respectively, 2 mm, 4 mm, 6 mm, 8 mm, and 10 mm. For the sensor head, the inner diameter, outer diameter, and thickness of all three permanent magnets are, respectively, 85 mm, 120 mm, and 10 mm. The permanent magnets are made of NdFe52 material and the coercivity is set to 955 kA/m. The remanence of them is 1.44 T, and the relative permeability along the axial direction is 1.21 [26]. The model is shown in Figure 3. To make a better simulation for a real situation, an air gap between the 2 half circles was set to 10 mm. The axial component and radial component of magnetic flux density are, respectively, extracted at a distance of 30 mm from the outer surface of the wire rope. Finally, MFL signals of both axial and radial components are collected as two characteristic curves shown in Figure 4a,b.

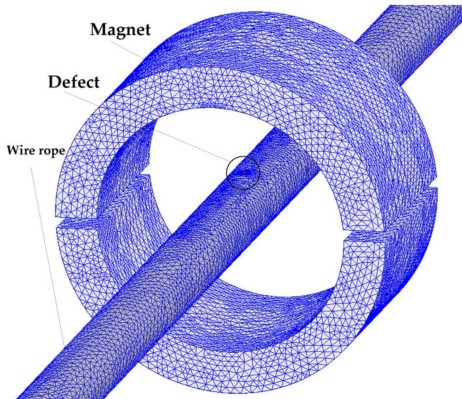

**Figure 3.** Mesh on the model.

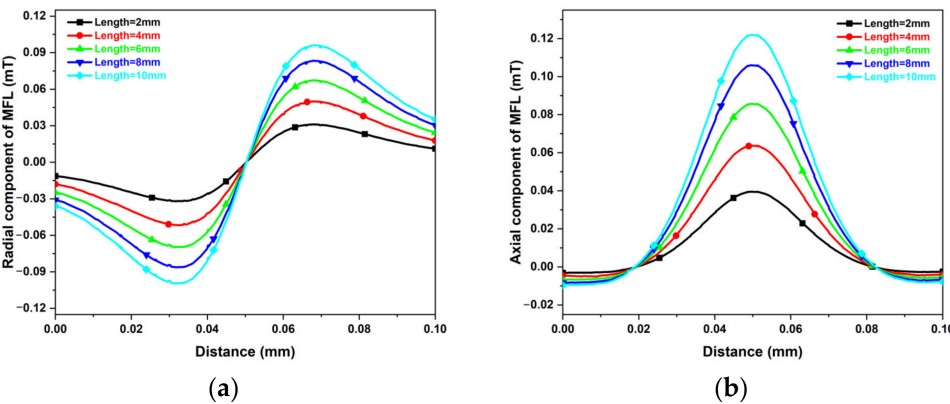

**Figure 4.** (**a**) Radial component of MFL; (**b**) Axial component of MFL.

The peak-peak value of the radial component and the peak-zero value of the axial component are, respectively, measured from the two curves above. To intuitively observe the relationship between the MFL signal and the length of the broken wire, Figure 5 is drawn below. In Figure 5, the B denotes the magnetic flux density of the leakage magnetic field, and the eigenvalue of the MFL signal is calculated from the peak-peak value of the radial component illustrated in Figure 4a.

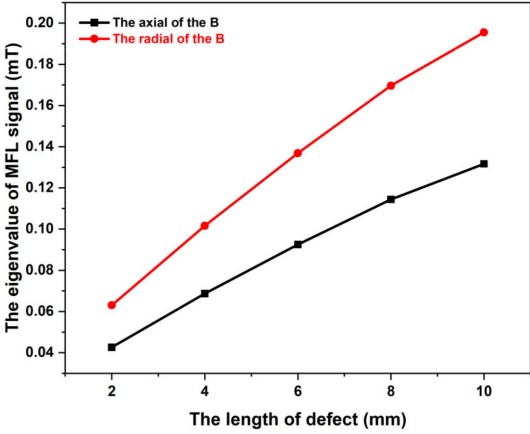

**Figure 5.** Relationship between the MFL signal and the length of the broken wire.

According to the above simulation analysis, MFL can be detected from the defect at a distance of 30 mm from the surface of the wire rope. Despite that, the amplitude of

the signal is weak. More specifically, useful information could easily be overwhelmed or vulnerable by noise from other resources.

### 2.3. Preliminary Experiment

Considering the MFL signal is weak at a distance of 30 mm, a preliminary experiment was done to verify the feasibility of large-lift-off detection using the existing equipment, which detects the signal at a distance of about 20 mm to 25 mm.

At the start, the existing sensor head used at a distance of 5 mm or less was improved. As shown in Figure 6, machining a groove whose deepness is 5 mm and width is 30 mm in the middle of the armature, as shown in Figure 6, aims to install magnetic sensors.

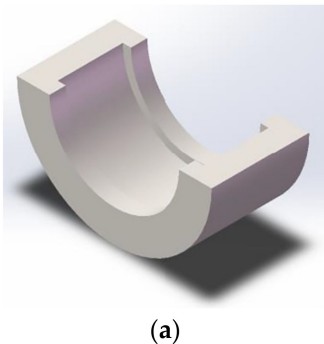 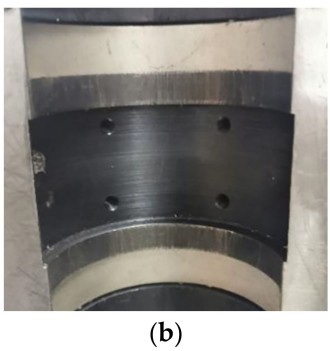

(**a**)        (**b**)

**Figure 6.** (**a**) 3D model armature; (**b**) The real armature.

The second change in the existing sensor head is to use wear-resistant sleeves of different sizes, as shown in Figure 7. To add up lift-off distance, different sizes of sleeves combination can make the sensor head eccentric to the detected wire rope.

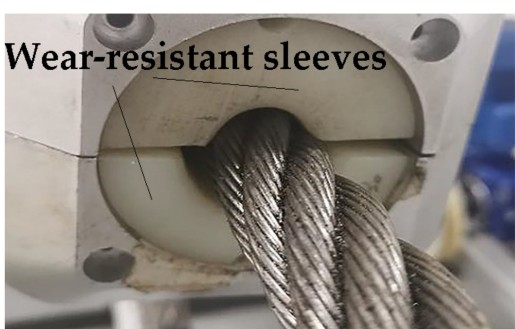

**Figure 7.** Wear-resistant sleeves.

Select the 24 mm diameter wire rope as the test object, make three different defects on it and collect the signal at the distance of 20 mm and 25 mm shown in Figure 8. The signals are shown in Figure 9a,b, and the spikes represent the MFL signal of three defects collected by coil sensors. Specifically, the detector was moved from left to right and returned. Consequently, the spikes from left to right represent the defects of a single broken wire, binding broken wire, and two broken wires, respectively. The results show that a large lift-off distance in collecting wire rope defect signals is feasible. Meanwhile, the size of the large-lift-off-distance sensor head is determined according to the previous experiment. The inner diameter, outer diameter, and width of the ring magnets are, respectively, 85 mm, 120 mm, and 10 mm. The actual sensor head is shown in Figure 10.

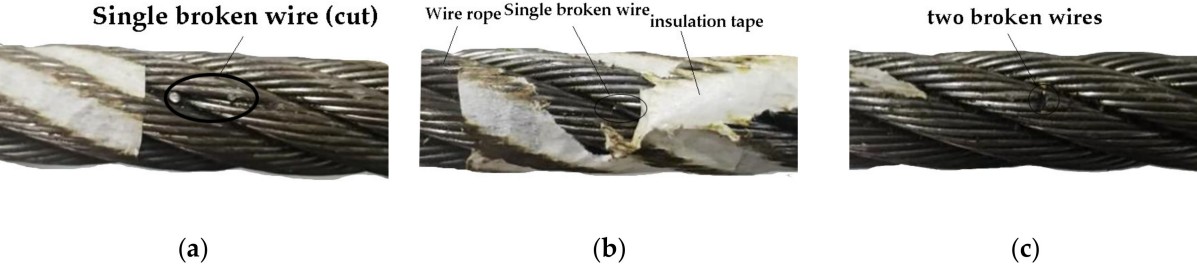

**Figure 8.** (**a**) Single broken wire; (**b**) Binding single broken wire; (**c**) two broken wires.

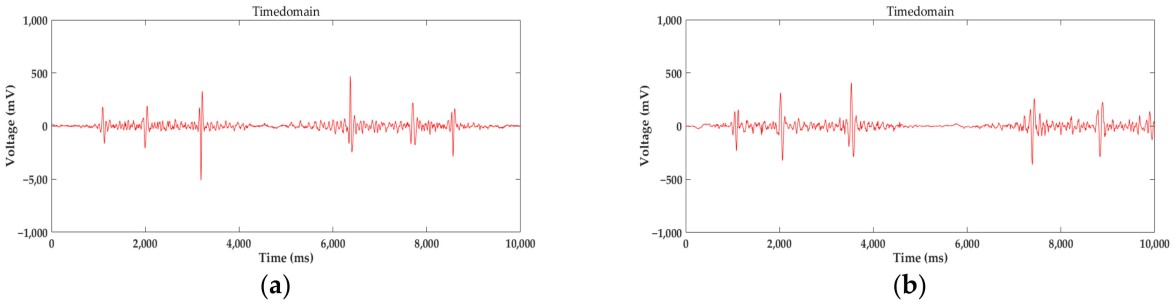

**Figure 9.** (**a**) 20 mm lift-off distance signal with the 24 mm diameter wire rope; (**b**) 25 mm lift-off distance signal with the 24 mm diameter wire rope.

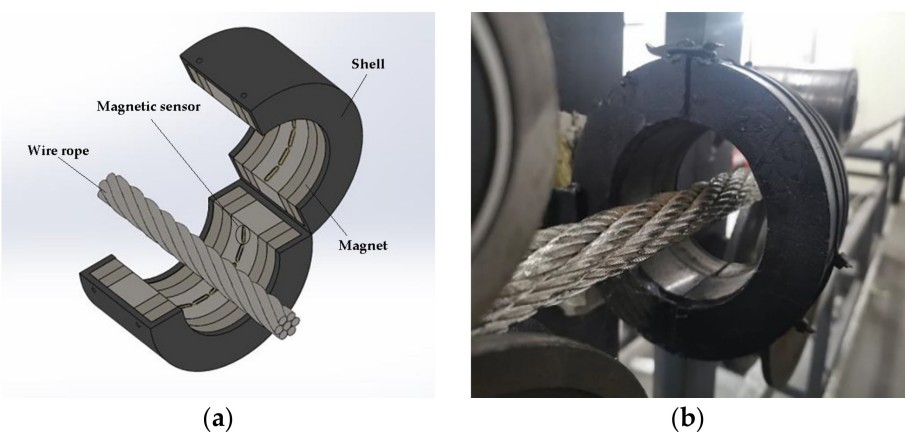

**Figure 10.** (**a**) 3D model sensor head; (**b**) Real sensor head.

## 3. Induction Coils

### 3.1. Coils Type

Induction coils have advantages such as low cost and endurance, so they are widely used in wire rope defect detection. Additionally, induction coils have many other forms, including coils made of enameled wire, flexible printed coils (FPC), and integrated inductance. And the three different kinds of coils shown in Figure 11 are tested in the next experiment.

To begin with, for the integrated inductance, due to its sensitivity being relatively low, they won't introduce too much noise, and it is usually applied in detection at a state of small lift-off distance. Thus, the signal of the defect at a large lift-off distance gets restrained by its low sensitivity.

Next, for the flexible printed coils, its sensitivity is a little bit higher than integrated inductance, because it can detect serious defects such as the irregular joining of wires or untwining of wires in strands. Nonetheless, a single broken wire signal cannot be distinguished from the sampling.

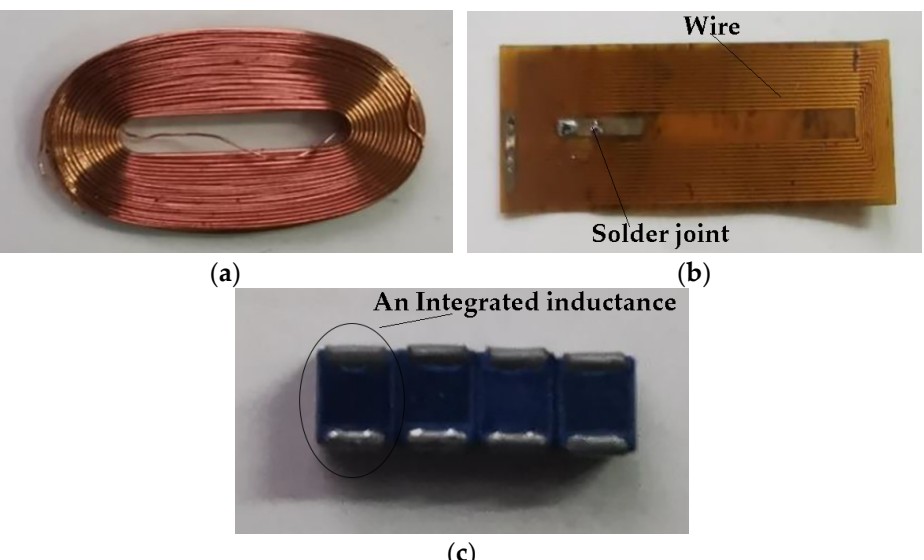

**Figure 11.** (**a**) Coils made of enameled wire; (**b**) Flexible printed coils; (**c**) Integrated inductance.

At last, the coils made of enameled wire have a high sensitivity leading to overwhelmed noise in the useful signal. Three different kinds of signals showed in Figure 12, where ID denotes induction, FPC denotes flexible printed coils, and Coil means coil made of enameled wire

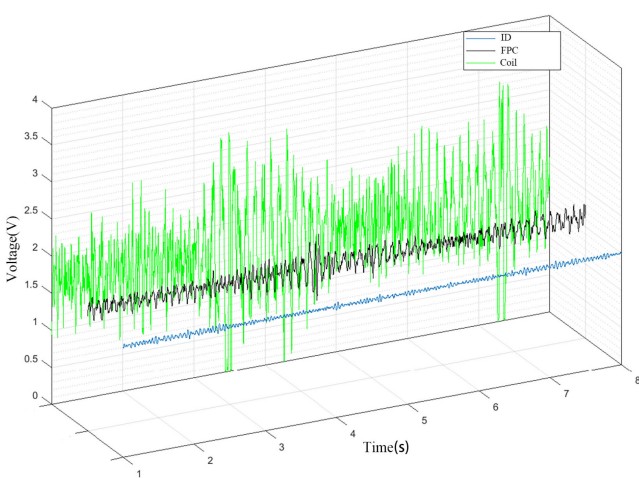

**Figure 12.** Signal collected by three kinds of coils.

According to the experiments above, all three different kinds of coils cannot get the distinguishable signal. Therefore, it is challenging but meaningful to find one kind of suitable sensor in a wire rope detecting system. The results also demonstrated that the Coil has the best sensitivity and could be used for large lift-off distance detecting.

### 3.2. Magnetic Sensor Array Coupling

Traditional main magnetic sensors array has three types. The first one is the entangled shown in Figure 13a. It's complicated and time-consuming to install or wind the coil at different working sites. The second one is the split with a one-row array shown in Figure 13b. At the state of small lift-off, the MFL signal can be detected with a good SNR. However, as the distance becomes larger, the energy excited by the wire rope defect's leakage magnetic flux declines exponentially. Hence the signal collected has low SNR under no sensors or signal processing conditions. The third one is to differential signal by laying on two rows of magnetic sensors shown in Figure 13c. The spacing between the

rows is equal to an integral multiple of spinning distance to match the theory of having the same strand MFL signal. After differential circuit processing, high SNR signal can be gotten. But unless the accuracy of the distance could be guaranteed, low SNR signal would occur. The signal is mixed with noise which is collected by two different arrays of the sensors. Especially, if the diameter of the wire rope is larger, the spinning distance will be farther. As a result, the axial length of the sensor head will be long and heavy, which has disadvantages in practical applications.

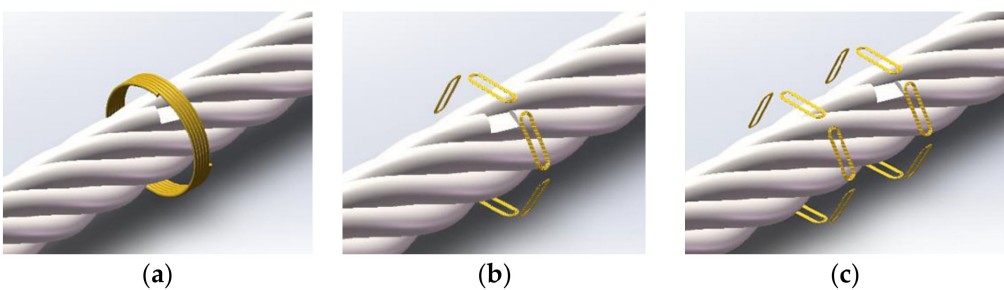

(**a**)   (**b**)   (**c**)

**Figure 13.** (**a**) Entangled coil; (**b**) Split with one-row array; (**c**) differential of two rows.

Using the traditional wire rope detecting methods, mainly hardware and software, to process the signal, the hardware circuit can filter the signal from its frequency. As shown in Figure 14, the sampling time is 10 s, and after processing the signal with noise at high speed by Fast Fourier Transform, the frequency range of the signal is almost limited to 10–30 Hz. Further, in the frequency domain analysis, different types of defects lead to different frequencies of defects. The noise could be removed through a filter and the defect signal would be received. However, only a frequency range can be gotten. The defect signal frequency and the noise frequency are so close, even mixed with each other, that we cannot distinguish the defect and noise correctly. Thus, the circuit filter cannot move the noise among the range. The defects were marked by a circle illustrated in Figure 14, and they are the same as experiments in Section 5.

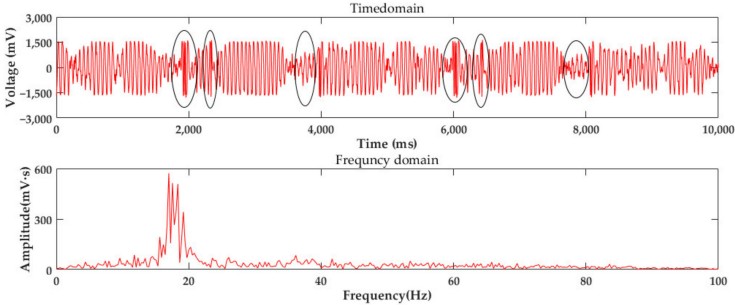

**Figure 14.** Time-Frequency diagram of the traditional type of magnetic sensors.

In conclusion, only using the traditional type of sensors array, the SNR of the signal picked is still low. Aiming to process the signal at the front end of the detecting system, exploring the parameters of coils and their array arrangement based on a large number of experiments is to be solved.

### 3.3. Coil Array

After many experiments, three types of coil array are proposed, including the side-by-side, the stacked, and the concentric shown in Figure 15. Build simulation model of wire rope detecting method based on open permanent magnetizer. In this model, the wire rope diameter was 22 mm, of 300 mm in length, and its material was X52, steel used for oil and gas pipeline transmissions, which is a medium grade in American Petroleum Institute 5L

specifications [27]. There was a defect including 2 mm (width) × 5 mm (length) × 2 mm (depth) on the surface of the wire rope. The lift-off distance between the sensor and the wire rope is 7 mm. The magnets' inner diameter, outer diameter, and axial length were respectively set to 85 mm, 120 mm, and 10 mm. Its material was NdFeB N52. The band area diameter was 24 mm and its length was 500 mm. The moving speed of the wire rope was 5 m/s, the time step was 0.2 s, and the whole process lasted for 24 ms.

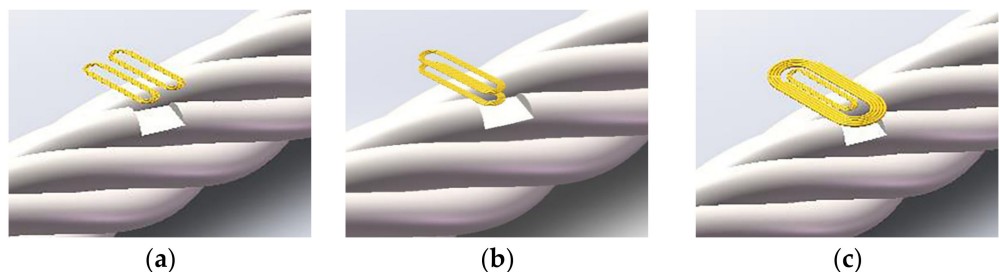

| (**a**) | (**b**) | (**c**) |

**Figure 15.** (**a**) Side by side; (**b**) Stacked; (**c**) Concentric.

Theoretical simulations were performed for the side-by-side arrangements of two induction coils shown in Figure 16a. Figure 16b schematically demonstrated a cloud map of magnetic flux density, and it is obvious that there is a leakage of magnetic flux around the defect leading to coil induced voltage. Finally, the output voltage attained is shown in Figure 17, and according to the analysis, it is preliminarily proved that the output electromotive force of different coupled arrays of coils is different. Furthermore, the concentric coil array can get the most significant output voltage amplitude, due to its low noise and wide signal band The stacked could easily lead to noise, and the stacked could be easily vulnerable to wire rope strands.

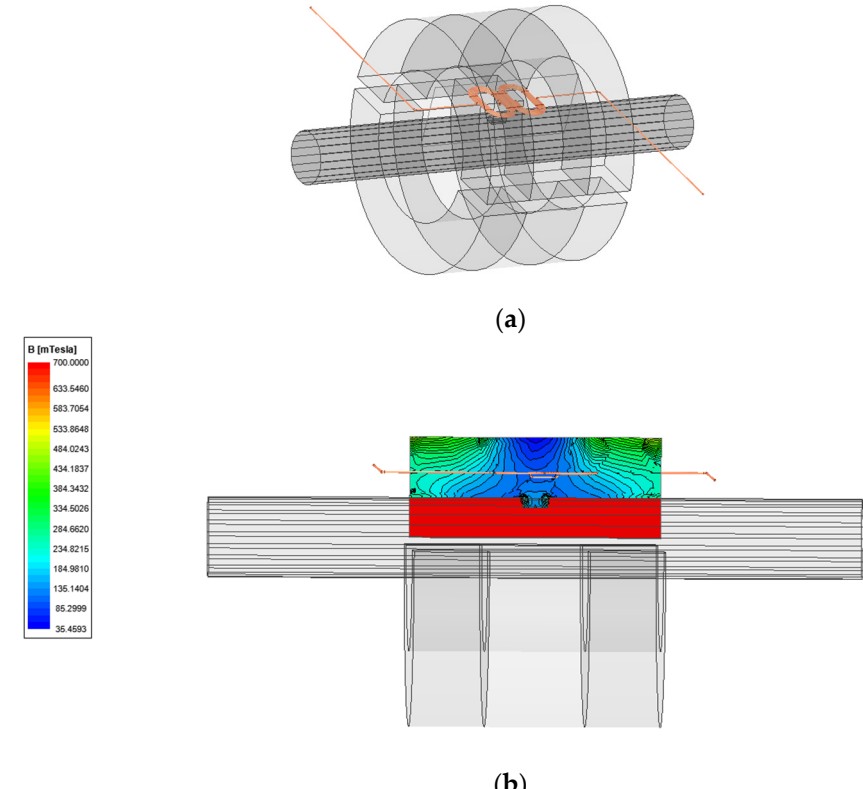

**Figure 16.** (**a**) 3D simulation model; (**b**) Cloud map of magnetic flux density.

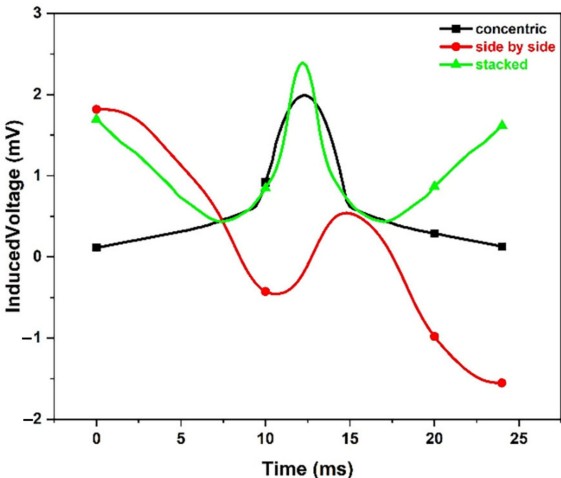

**Figure 17.** The output voltage of three types of coil arrays.

### 3.4. Coil Structure Parameters

After researching coil arrays, coil structure parameters, shown in Figure 18, are also essential to be determined. In Figure 18, A, B, and C are, respectively, the width of the air gap, inner coil, and outer coil. Different turns of a coil or different wire diameters can be studied by controlling for a single variable A, B, or C. Moreover, it can be considered from the following aspects.

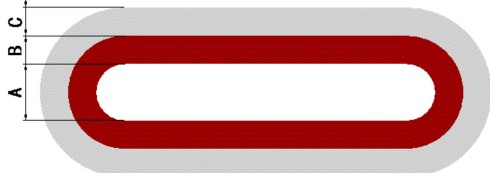

**Figure 18.** Coil structure parameters.

The first one is the reasonable wire diameter. Small wire diameter means the small volume of the sensor. In this way, the number of turns and the length of the wire could be added to improve the sensor's sensitivity. Nevertheless, the coil resistance increases as the diameter get smaller, increasing the signal loss and the complexity of the welding process.

The next is to choose suitable inductance because the more inductance, the more output voltage. To improve the inductance, the number of turns and cross-section area of wire should be increased. Whether increasing the inductance or reducing the resistance value, suitable coil parameters can improve the quality factor of the coil.

## 4. Analog Circuit

### 4.1. Circuit Design

In detecting wire rope, the MFL signal collected by the sensors is tiny, but the intensity of the magnetic field decreases exponentially as the lift-off distance linearly increases. Aiming to amplify the weak signal and remove noise signal, a signal processing circuit to perform accurate filtering was proposed in this section. Considering the application background of steel wire rope is mostly mine and ocean, where the working conditions are harsh, some fundamental requirements have been proposed, including the following:

- High input impedance;

Signal source internal resistance can reach 100 $\Omega$. If the loss on the signal source is required to be less than 1%, then the input impedance is required to be more than 10 k$\Omega$.

- High voltage gain;

The signal collected is feeble at the considerable lift-off detection distance (>30 mm), even less than millivolts. Therefore, the circuit amplification gain should reach more than 1000 times.

- Single power supply;

Due to the dual power supply with the negative voltage supply, additional voltage converter and coupling capacitors are required, leading to the complex design of the circuit and more noise sources. To simplify the structure of the circuit and reduce the size of the signal processing module, a single power supply has been adopted, which is also convenient to install and arrange.

- Low noise;

Owing to the harsh environment with many kinds of noise, electronic components must have high stability to avoid interference from various noises. Besides, parameter characteristics of components do not change significantly with temperature.

- Proper bandwidth;

According to many experiments, the frequency range of the damage signal of the wire rope under normal speed conditions is below 200 Hz. Even though the wire rope is running at a higher speed, the frequency range of the defects signal is still less than 500 Hz. As a consequence, amplification of low-frequency signals will be the focus.

According to the targets above, the circuit structure is divided into three functional modules. The first one is weak signal amplification. The second one is a low pass filter. The third one is the high pass filter. The operational amplifier circuit is composed of OPA2188, shown in Figure 19. OPA2188 was chosen from National Instrument as an operational amplifier, and its main features include low drift voltage, low noise, zero drift, and high CMRR. Its pin diagram is shown in Figure 20 In means input and out means output. It is a dual operational amplifier. V means DC voltage.

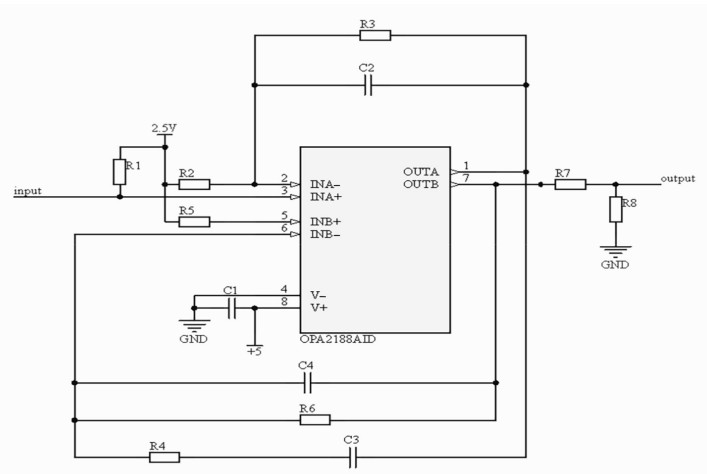

**Figure 19.** Operational amplifier composed of OPA2188.

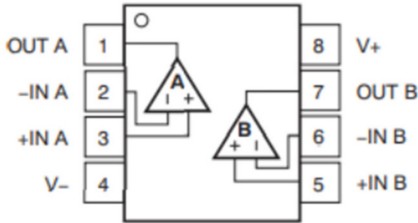

**Figure 20.** OPA2188 pin diagram.

When designing the operational amplifier, a two-stage amplification layout was adopted. The amplification factor $A_1$ of the first-stage non-inverting amplifier circuit is

$$A_1 = 1 + \frac{R_3}{R_1} \tag{1}$$

$R_1$ is the balance resistor in the first-stage amplifier circuit, and it is used to provide a DC path to the non-inverting input. $R_2$ forms a loop with GND to generate base current. The amplification factor $A_2$ of the second-stage inverting amplifier circuit is

$$A_2 = -\frac{R_6}{R_4} \tag{2}$$

Amplification of every stage was set to 100, so the overall amplification is 10,000. $R_5$ is the balance resistor in the second-stage inverting amplifier, and it is used to provide suitable static bias and eliminate the effect of base current on the output voltage. Meanwhile, the operational amplifier was supplied by a DC bias voltage of 2.5 V while using a single power supply. Considering the signal collected is altering voltage, the output voltage swing around 2.5 V. On the one hand, the negative voltage collected can be output correctly. On the other hand, a more excellent dynamic range can be achieved.

Additional active or passive filters are no longer required to remove the high-frequency noise, but existing feedback resistors in parallel with feedback capacitors are utilized to form active filters. Connect the feedback capacitor between the two stages to compose a second-order low pass filter. The cut-off frequency is

$$\text{fH} = \frac{1}{2\pi C R_f} \tag{3}$$

OPA2188 has been chosen for the operational amplifier because of its high Common Mode Rejection Ratio (CMRR), and OPA2188 is a dual operational amplifier.

### 4.2. Circuit Simulation Analysis

The designed circuit is simulated and analyzed. Most defects' frequency is concentrated at about 20 Hz at the wire rope's average running speed (1–2 m/s). The frequency band was set at 0–300 Hz, and the central frequency was 20 Hz. According to the filter's cut-off frequency, suitable parameters of resistors and capacitors were chosen. The frequency response curve is shown in Figure 21. The central frequency is 21 Hz, and the amplification factor is 1900. However, the amplification factor decreases to 150 at 300 Hz.

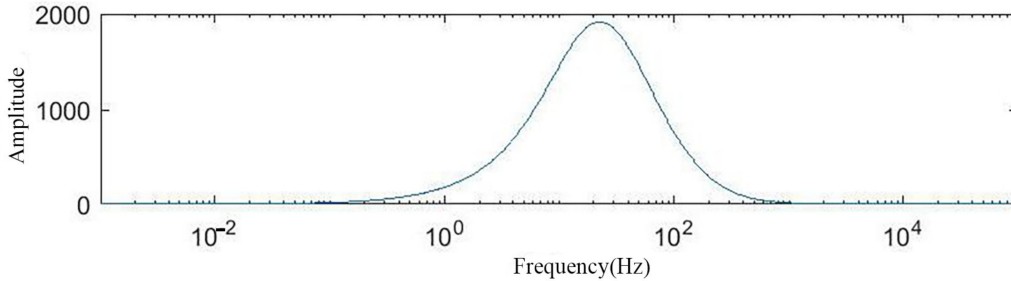

**Figure 21.** Amplifier circuit frequency response characteristics.

### 5. Experiments

To determine the best parameters of coil sensors mentioned in Section 3, a series of experiments were carried out to preliminary study the SNR of the signal collected by coil sensors. This proceeds in two stages: the first was to focus on the best coil array, and the second was to acquire the best structure parameters of coils. Based on the experiment results, different influences of parameters were investigated and discussed in detail. All of

the signals collected are shown below, including the three types of defects; a circle marks the single broken wire signal among the defects.

### 5.1. Experiment Arrangement

A horizontal high-speed rotary test bench was constructed to implement the detection of wire rope at a considerable lift-off distance. During the test, the motor drives the main shaft, and the detected wire rope moves with the pulley. The moving speed of the wire rope is controlled by a variable-frequency drive (VFD), and the maximum speed can reach 10 m/s. The detecting system is shown in Figure 22, consisting of a horizontal high-speed rotary test bench, large lift-off detecting probe, signal extension cable, and data acquisition and processing system. At the same time, the detecting probe is shown in Figure 23 and was fixed on the adjacent pillar to guarantee a certain lift-off distance. The magnetizer was preliminarily determined in Section 2, which has enough intensity of magnetizing at a considerable lift-off distance. By putting non-magnetic material between the wire rope and magnets, the distance between the defect and the coil sensors can be changed.

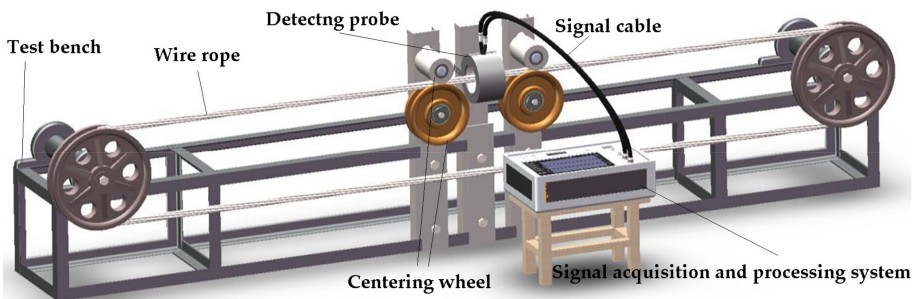

**Figure 22.** Detecting system.

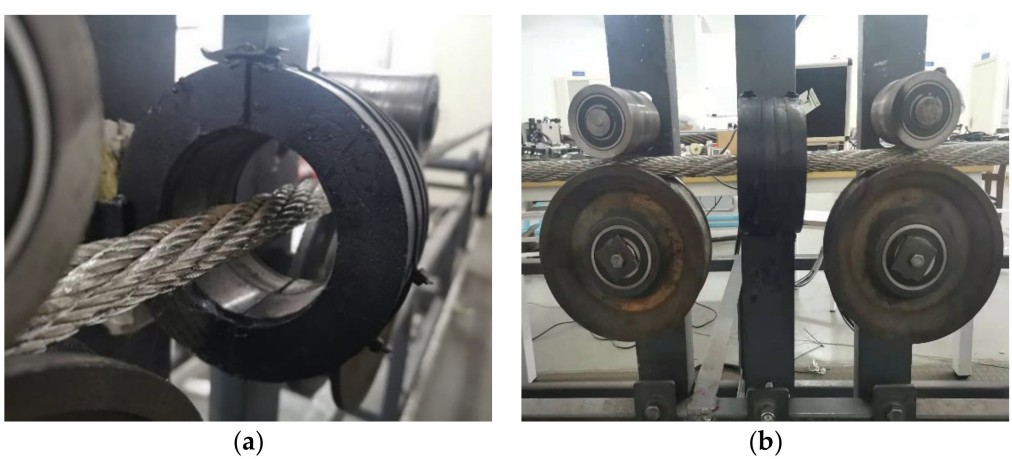

**Figure 23.** (**a**) Front of detecting probe; (**b**) Side of detecting probe.

In these experiments, additional damages were designed, and their signals were acquired. The diameter of the detected wire rope is 22 mm. What is more, the inner diameter of the magnetizer is 85 mm; considering the thickness of the sensors, the lift-off distance can reach 30 mm. There are three primary defects on the whole wire rope; one untwine of wires in the strand, one irregular joining of wires, and a single broken wire, whose length is 1 mm, bundled by insulation tape shown in Figure 24. Besides, the Printed Circuit Board (PCB) designed in Section 4 shown in Figure 25 was tested and connected to the existing wire rope magnetic flux leakage detection system in the experiments. In Figure 25, the left side is the amplifying circuit board, and the right is the lower computer STM32.

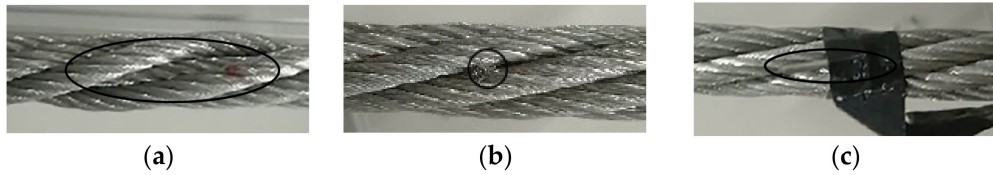

**Figure 24.** (**a**) Untwine of wires in strand; (**b**) Irregular joining of wires; (**c**) Binding single broken wire.

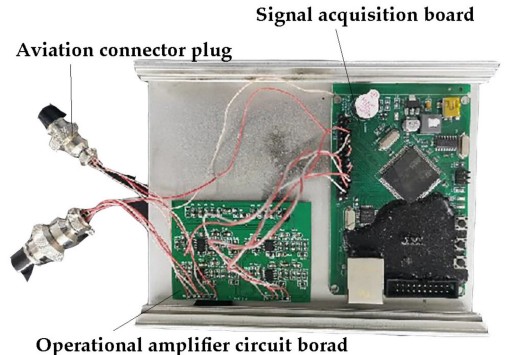

**Figure 25.** PCB.

### 5.2. Effect of Coil Arrays

As shown in Figures 26 and 27, the running speed of the wire rope is 5 m/s, and the lift-off distance is 15 mm. We analyzed the data collected from three types of coil arrays with the magnetizer mentioned above for this study. Comparing the signal waveforms, the results show that the coil arrays arranged side by side acquire much noise. Comparatively, the stacked and the concentric can collect the single broken wire defect, represented by the minor spike. More specifically, the SNR of the signal collected by the concentric coil array is better than the stacked, and the defect signal cannot be distinguished from the side-by-side sensor.

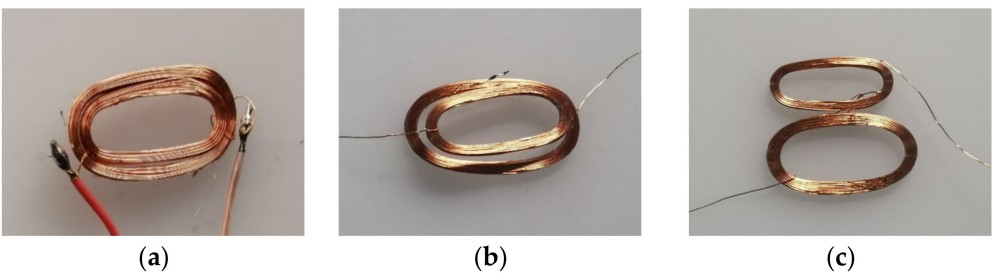

**Figure 26.** (**a**) Concentric; (**b**) Stacked; (**c**) Side-by-side.

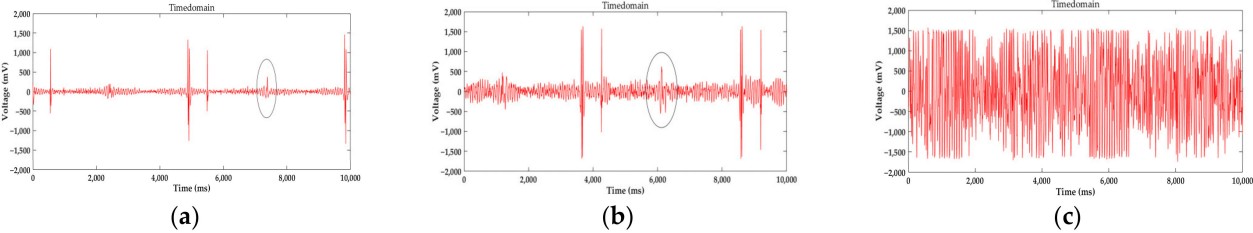

**Figure 27.** (**a**) Concentric; (**b**) Stacked; (**c**) Side-by-side.

Meanwhile, all the coil arrays in the experiment are connected differentially. To prove the effectiveness of the differential connection, under the situation that the running speed is 10 m/s, Figure 28 demonstrates that the differential connection can improve SNR and

reduce noise, which is caused by the swing signal and strand signal. It can be concluded that the differential connection can reduce noise because the single broken wire defect signal is easily distinguished. In all of the experiment signal figures, the enormous amplitude and the smallest amplitude, respectively represent untwine of wires in a strand and the single broken wire. The defect of irregular joining of wires is located between the two defects.

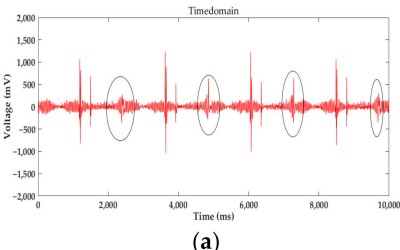
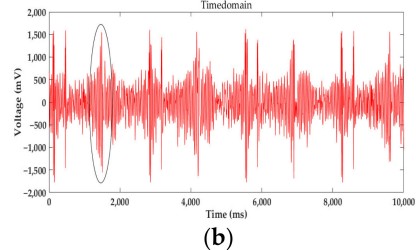

(**a**)    (**b**)

**Figure 28.** (**a**) Differential connection; (**b**) Non-differential connection.

### 5.3. Effect of Coil Structure Parameters

Firstly, the effect of the number of coil turns on the output signal was explored in the experiment. The experimental conditions are the Φ22 mm (Φ denotes diameter) wire rope as shown in Figure 24, the wire rope running speed was 6 m/s, and the lift-off distance was 15 mm. The experimental results are shown in Figures 29 and 30. There are three different kinds of coil arrays compared. The first is A = 2, B = 1, C = 1; the second is A = 2, B = 1, C = 3 and the third is A = 2, B = 3, C = 1. A, B, and C are schematically demonstrated in Figure 18. It can be known by analyzing the result that as the number of turns of the magnetic sensitive coils increases, the SNR of the output signal decreases. In this experiment, to ensure that the receiving area of the coil remains unchanged, the total area of differential coil sensors should keep consistent. In reality, this similar problem was studied by Zhang [28], whose research papers say that the closer to the center of the enameled wire coil, the smaller size of the coil. This reason leads to small inductance and a low-quality factor. Thus, it can be concluded that the SNR of the coil array with many inner turns is lower than that of the coil array with many outer turns. The more turns, the more sensitive the coil sensors are, leading to collecting much noise during the test. However, too few coil turns will also affect the signal acquisition, which could lead to almost no signal, resulting from low sensitivity.

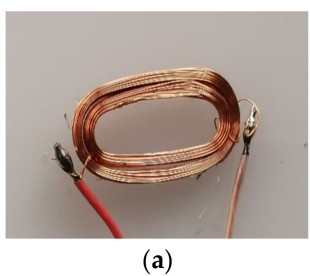
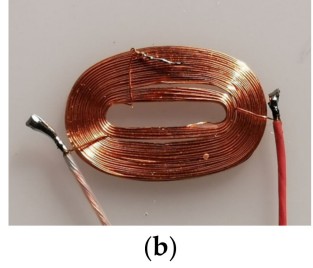
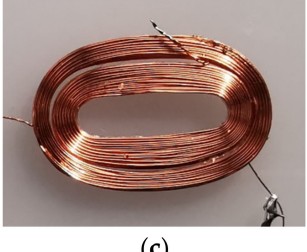

(**a**)    (**b**)    (**c**)

**Figure 29.** (**a**) A = 2, B = 1, C = 1; (**b**) A = 2, B = 1, C = 3; (**c**) A = 2, B = 3, C = 1.

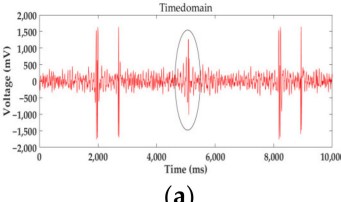
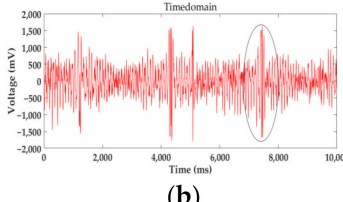
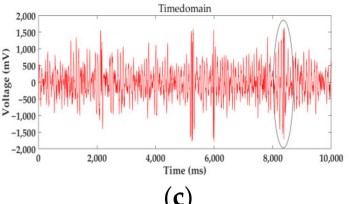

(**a**)    (**b**)    (**c**)

**Figure 30.** (**a**) A = 2, B = 1, C = 1; (**b**) A = 2, B = 1, C = 3; (**c**) A = 2, B = 3, C = 1.

Secondly, the relationship between the coil magnetic flux area and the output signal is explored. There are four different kinds of coil arrays compared shown in Figure 31. The only variable is the area of the air gap, A, and A ranges from 2 to 5. Experimental conditions were the same as in the above experiment, and the only difference was the lift-off distance. The lift-off distance in Figure 32a–d is 15 mm, and in Figure 32e–h is 30 mm.

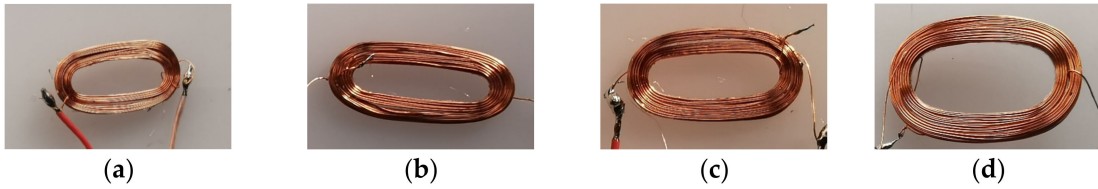

**Figure 31.** (**a**) A = 2, B = 1, C = 1; (**b**) A = 3, B = 1, C = 1; (**c**) A = 4, B = 1, C = 1; (**d**) A = 5, B = 1, C = 1.

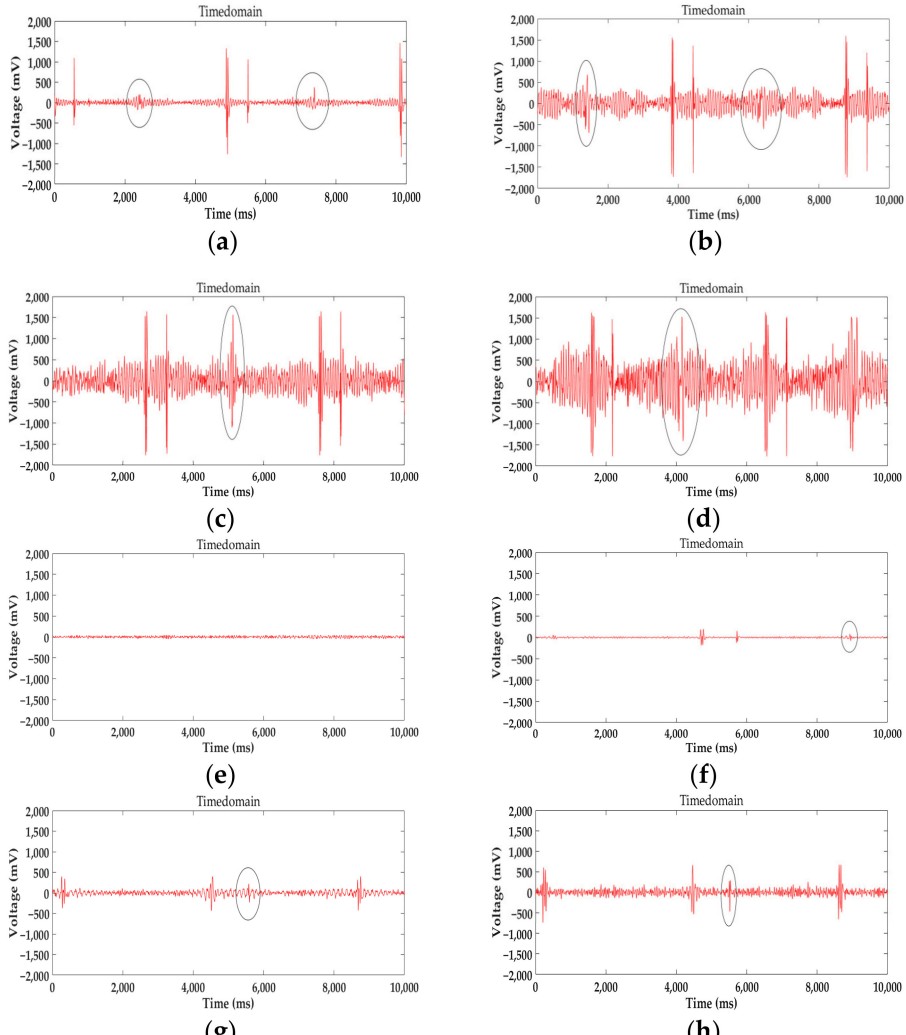

**Figure 32.** (**a**) A = 2, B = 1. C = 1; (**b**) A = 3, B = 1, C = 1; (**c**) A = 4, B = 1, C = 1; (**d**) A = 5, B = 1, C = 1; (**e**) A = 2, B = 1. C = 1; (**f**) A = 3, B = 1, C = 1; (**g**) A = 4, B = 1, C = 1; (**h**) A = 5, B = 1, C = 1.

By analyzing Figure 32, the larger area of magnetic flux, the more sensitive the coil is, and the sensor can pick up a lot of weak signals so that the single broken wires can be distinguished easily. Even though combined with a–d, as the area of magnetic flux increases, the SNR will decrease. The reason for the phenomenon is that the sensor has greater sensitivity causing more noise to be captured. It shows that the magnetic flux area is

not as large as possible, and it needs to be coupled with the lift-off distance. To improve the ability to collect signals, the magnetic flux area of the coil with a lift-off distance of 30 mm should be significantly analyzed from Figure 32e–h. The area of the coil sensor should be relatively small with a lift-off distance of 15 mm, which reduces the acquisition of noise signal and improves the SNR of the output signal. Together, the present findings confirm that the sensitivity of the coil sensors should couple with the different magnetic fields to gain an ideal and clear detection signal. The implications of these findings are discussed in Section 6.

*5.4. Effect of Magnetic Material*

In this experiment, magnetic materials mainly refer to soft magnetic materials. The soft magnetic material can achieve maximum magnetization with the minimum external magnetic field. In the experiment, permalloy was used as the magnetic material. Permalloy is a Fe-Ni-based soft magnetic alloy with extremely high magnetic permeability under a weak magnetic field. Thus, the magnetic flux will accumulate when the permalloy is added to the coils, as shown in Figure 33. The sensitivity of the sensor can be improved.

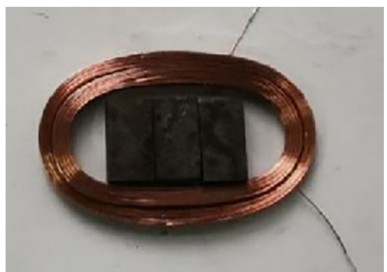

**Figure 33.** Coil array with permalloy.

In the experiments mentioned above, the running speed of the detected wire rope is relatively high (≈5 m/s). According to the Faraday law of electromagnetic induction, the wire rope's slow speed leads to a weak output voltage. Based on extensive experimental verification, in most applications, such as slow speed or considerable lift-off distance, permalloy can significantly improve the detection ability of magnetic sensitive sensors to magnetic leakage flux of the defects. As shown in Figure 34, the experimental condition is that the lift-off distance is 30 mm at a relative speed of 2 m/s. The peak to peak value of the single broken wire signal was increased from 275 mV to 466 mV. We obtain good results with this simple method.

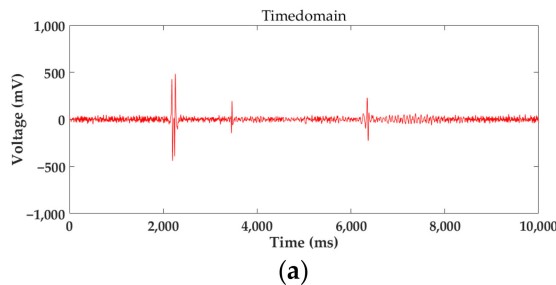 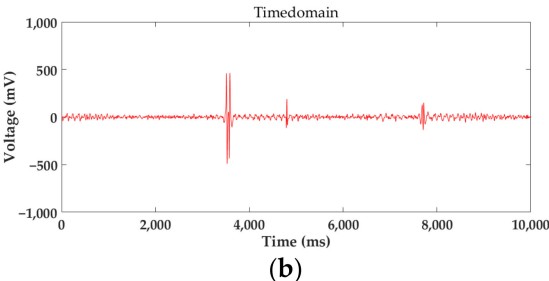

(a)                         (b)

**Figure 34.** (**a**)Coils array with permalloy; (**b**) Coils array with air.

In summary, the testing device in the more than 30 mm detecting distance adopts the coil whose size is A = 5 mm, B = 1 mm, C = 1 mm, and the inner space is full of permalloy.

## 6. Discussion

In this article, a more than 30 mm detecting distance testing device for wire rope was proposed to solve the problem that the inner surface of the sensor head is close to the wire

rope leading to the detector scraping off the surface oil of the wire rope or stuck by cut wire in the course of MFL detection. The open permanent magnetizer was determined. Moreover, that there is still magnetic leakage flux at a large lift-off distance (>30 mm) was theoretically analyzed using a 3-D simulation model. Furthermore, the detection signal at the 20 mm and 25 mm distances were collected. The feasibility of detecting at a considerable lift-off distance was verified, and the magnetizer was designed. Subsequently, the output signal characteristics of the different types of coils, the influence on the output signal of different kinds of the coupled array, and the different structure parameters of coils were discussed. In addition, the analog circuit was designed to achieve the goal of amplifying the weak signal and removing the noise signal. According to the weak signal output characteristics of the signal source and noise types, an analog signal processing method based on non-inverting amplification was proposed, the circuit was designed, and the relevant electronic components were selected. Finally, by retrofitting the magnetizer and combining magnetic sensing of weak magnetic field and analog processing method of weak signal, a horizontal high-speed test bench was constructed to research the best type of coil arrays and structure parameters. Besides, for the relatively low speed of the detected wire rope, soft magnetic material, namely permalloy, was used to improve the sensitivity of the coil sensors. The testing device in the more than 30 mm detecting distance was determined.

By comparing the results from the experiments above, under the same diameter wire rope, the same magnetizer, and the same detection lift-off distance conditions, the detection ability of sensors is focused. A new magnetizer is proposed based on an open permanent magnetizer, and a PCB to process signal is designed. Extensive results carried out show that this method can be used to detect defects at a distance of 30 mm.

Notably, different coil arrays lead to different coil sensitivity. This implies that the sensitivity of sensors has something to do with sensors' structure and mutual coupling between coils. We put forward that the different parameters of differential coil sensors result in different sensitivities. Results in Sections 5.2 and 5.3 indicate that a higher sensitivity sensor is required at the large lift-off distance detection. By comparison, sensors' sensitivity should be decreased when the lift-off distance is only less than 15 mm or the wire rope speed is comparatively slow. We confirm that a differential coil sensor can decrease the sensitivity but increase SNR. The present findings confirm that more turns, larger detection areas, or thinner coil wire diameters will add up coils' sensitivity. It is also interesting to note that soft material can add up sensor sensitivity. It means that some magnetic materials may improve the detection ability, and our future work will include finding any material to make crafts of magnetic sensors better. In addition, the simulation result in Figure 17 is also verified through experiments in Section 5.2. We discovered that the concentric array has the lowest background noise, and the stacked array has a larger signal amplitude but easily causes the noise. In contrast, noise collected by the side-by-side array and caused by the rope strand is similar to the defect MFL signal, leading to the defect signal cannot be distinguished.

The current work is sufficient to point out that the defect can be detected at a large lift-off (>30 mm) distance using the MFL method. Experiments with the characteristic of large lift-off distance and high-speed wire rope (>2 m/s) were performed for the first time. We cannot deny the presence of some sample selection biases because the rules of other situations are complex. Our data address different speeds from 1 m/s to 10 m/s, and many different A, B, and C sizes are also explored. However, without data processing statistically and theoretical support, a further conclusion is not drawn yet. We just pick up part of them, which correspond with Faraday's Law, to reveal typical rule which is also applied to other circumstances, such as different wire rope running speeds and different types of enameled wire coils.

Because of the lack of theoretical support, we decided not to investigate other kinds of coil sensors, which have different shapes, wire diameters, the number of turns, and other similar parameters. Owing to this potential limitation, we treat only one type of coil, whose wire diameter is 0.12 mm, and its thickness is 1 mm. The parameters of A, B, and C are

changed by increasing or decreasing the number of turns. Another limitation of this study is the simulation process. A major source of the limitation is not clearly understanding the magnetic field coupling process; it is just a preliminary 3-D model simulation of the sensor's output voltage. Lots of details must be studied and analyzed clearly in the simulation program. Overall, our method is the one that obtained the most popular results that the signal collected should consider the parameters of wire rope, magnetic field, the sensitivity of sensors, and even the detection distance and the moving speed.

## 7. Conclusions

This article proposes a testing device for wire rope based on an open magnetizer at a 30 mm detecting distance. The magnetic characteristic of the magnetizer is verified by finite-element simulation, and the feasibility of detecting at the considerable lift-off distance is initially verified by experiment. Induction coils are analyzed from the type, the array, and the structure parameter. An analog circuit is used to eliminate noise at different frequencies. The experiment compares different groups of coils with the constructed system. Comparing different coil arrays and coil structure parameters, the best kind of coil sensor is proposed to detect at a 30 mm lift-off distance. For the relatively low speed of the detected wire rope, permalloy was used to improve the sensitivity of the coil sensors to collect some weak damage signals.

However, in this article, the quantitative relationship between the relevant parameters of the magneto-sensitive element and applicable conditions, including the amount of magnetization, the diameter of the wire rope, and the lift-off distance, need to be further explored. It is only the empirical data obtained based on many experiments. If the problem can be further studied, it will help to improve the detection performance of the wire rope magnetic flux leakage detection probe.

The future work will focus on the quantitative relationship between magnetic sensors and magnetic leakage flux coupling, including theoretical analyses and finite-element simulations. How to enlarge the detective area and distance of coil sensors will also be investigated.

**Author Contributions:** Conceptualization, Y.S. and M.L.; methodology, C.Z.; software, X.J. (Xiaotian Jiang); validation, M.L., C.Z. and X.J. (Xiaoyuan Jiang); formal analysis, X.J. (Xiaoyuan Jiang) and M.L.; investigation, C.Z. and M.L.; resources, Y.S.; data curation, M.L.; writing—original draft preparation, M.L.; writing—review and editing, Y.S.; visualization, M.L.; supervision, L.H.; project administration, R.L.; funding acquisition, R.L. All authors have read and agreed to the published version of the manuscript.

**Funding:** This research was funded from Research on Operation and Maintenance Technology of Balance Weight Wire Rope programme, Changjiang River Administration of Navigational Affairs, MOT, grant number SXHXGZ-2021-2.

**Acknowledgments:** This work was enabled through Wuhan Yuyuan Detection LLC for the test bench established and data collected.

**Conflicts of Interest:** The authors declare no conflict of interest.

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
