# Peer review of "The Research of 30 mm Detecting Distance of Testing Device for Wire Rope Based on Open Magnetizer"

_applsci, doi:10.3390/app12104829_

Round 1

Reviewer 1 Report

Dear Authors,

Please find comments in the attached PDF file as sticky notes.

Reviewer 2 Report

The authors used finite element analysis was used to verify if the MFL signal exists at the large lift-off. They combined magnetic sensing and coupling and weak analog signal processing method. My constructive criticisms of the article are itemized below:

  1. How did the authors obtain the material parameters used in the FEM analysis?
  2. Superficial comments about the computational model. It is not clear to me how the authors apply this method.
  3. Regarding the FEM model, the paper does not report any aspect associated with the convergence of the FE model. The convergence is a critical issue, both in terms of the quality of the solution and its ability to go beyond global displacements and quantify internal stress resultants and the corresponding stresses, and in terms of the computational load associated with the analysis.
  4. The experimental setup must be better explained. How many structures are analyzed for each condition? Also, how the authors treat the aliasing and leakage phenomena?
  5. In figures 27, 28, 30, 32 and 34, the authors nee to increase the size of the axes.
  6. The limitations must be discussed by the authors and to the related need of further work.

Round 2

Reviewer 1 Report

Dear Authors,

After the major revision the manuscript became much more understandable. Thanks for taking into account all the comments. The suggestions for minor corrections you can find in the attached PDF file as sticky notes.

Author Response

Line 56-66 in red was added to supplement the background and relevant reference of large lift-off MFL method. 

Line 629-632 & 636-641 in red was added to discuss experiments and simulation results in more detail.

5 references have been added.

Reviewer 2 Report

The authors have satisfactorily addressed all the queries raised by the reviewer.

Author Response

Response to Reviewer 2 comments

Point 1: the introduction can be improved.

Response 1: Line 56-66 in red was added to supplement the background and relevant reference of large lift-off MFL method. 

Point 2: the results presentation can be improved.

Response 2: Line 629-632 & 636-641 in red was added to discuss experiments and simulation results in more detail.